# Investigation of UTR Variants by Computational Approaches Reveal Their Functional Significance in *PRKCI* Gene Regulation

**DOI:** 10.3390/genes14020247

**Published:** 2023-01-18

**Authors:** Hania Shah, Khushbukhat Khan, Yasmin Badshah, Naeem Mahmood Ashraf, Maria Shabbir, Janeen H. Trembley, Tayyaba Afsar, Ali Abusharha, Suhail Razak

**Affiliations:** 1Department of Healthcare Biotechnology, Atta-Ur-Rahman School of Applied Biosciences, National University of Sciences and Technology, Islamabad 45860, Pakistan; 2School of Biochemistry and Biotechnology, University of the Punjab, Lahore 54590, Pakistan; 3Minneapolis VA Health Care System Research Service, Minneapolis, MN 55417, USA; 4Department of Laboratory Medicine and Pathology, University of Minnesota, Minneapolis, MN 55455, USA; 5Masonic Cancer Center, University of Minnesota, Minneapolis, MN 55455, USA; 6Department of Community Health Sciences, College of Applied Medical Sciences, King Saud University, Riyadh 11433, Saudi Arabia; 7Department of Optometry, College of Applied Medical Sciences, King Saud University, Riyadh 11433, Saudi Arabia

**Keywords:** 3′ UTR, SNP, non-coding region, 5′ UTR, PRKCI, miRNA

## Abstract

Single nucleotide polymorphisms (SNPs) are associated with many diseases including neurological disorders, heart diseases, diabetes, and different types of cancers. In the context of cancer, the variations within non-coding regions, including UTRs, have gained utmost importance. In gene expression, translational regulation is as important as transcriptional regulation for the normal functioning of cells; modification in normal functions can be associated with the pathophysiology of many diseases. UTR-localized SNPs in the PRKCI gene were evaluated using the PolymiRTS, miRNASNP, and MicroSNIper for association with miRNAs. Furthermore, the SNPs were subjected to analysis using GTEx, RNAfold, and PROMO. The genetic intolerance to functional variation was checked through GeneCards. Out of 713 SNPs, a total of thirty-one UTR SNPs (three in 3′ UTR region and twenty-nine in 5′ UTR region) were marked as ≤2b by RegulomeDB. The associations of 23 SNPs with miRNAs were found. Two SNPs, rs140672226 and rs2650220, were significantly linked with expression in the stomach and esophagus mucosa. The 3′ UTR SNPs rs1447651774 and rs115170199 and the 5′ UTR region variants rs778557075, rs968409340, and 750297755 were predicted to destabilize the mRNA structure with substantial change in free energy (∆G). Seventeen variants were predicted to have linkage disequilibrium with various diseases. The SNP rs542458816 in 5′ UTR was predicted to put maximum influence on transcription factor binding sites. Gene damage index(GDI) and loss of function (o:e) ratio values for PRKCI suggested that the gene is not tolerant to loss of function variants. Our results highlight the effects of 3′ and 5′ UTR SNP on miRNA, transcription and translation of PRKCI. These analyses suggest that these SNPs can have substantial functional importance in the PRKCI gene. Future experimental validation could provide further basis for the diagnosis and therapeutics of various diseases.

## 1. Introduction

Protein kinase C (PKC) family members are essential elements of signaling pathways that play a central role in the regulation of survival, proliferation, migration, cell cycle, and polarity [1]. The atypical members of this protein kinase family are structurally and functionally unique from other family members and do not require calcium or diacylglycerol (DAG) for catalytic activation [2]. These atypical members also do not function as cellular receptors [3]. Protein kinase C iota is a member of the atypical class encoded by the PRKCI gene located on chromosome 3. The catalytic activation of PKCɩ occurs through phospholipids, including phosphatidylinositol 3,4,5 triphosphate interactions with the Phox and Bem1 (PB1) domain of the N-terminal regulatory region [4]. PKCɩ contributes to tumorigenesis by metamorphosed growth and survival of cancer cells [5]. High expression of PKCɩ has been found in human chronic myelogenous leukemia (CML). Genetic interruption of PKCɩ either via dominant negative mutant of PKCɩ or by expression of anti-sense construct increases the sensitivity of the cells towards chemotherapy provoked cell death by blocking alteration caused by Bcr-Abl [6]. PKCɩ endorses colon tumorigenesis both in vitro and in vivo, increased expression was found in transgenic mice after developing preneoplastic lesions in the colon [7]. PKCɩ has been found to confer oncogenic Ras signaling [8]. It has been linked with Ras-intervened transformation in epithelial cells of intestine in rats that can cause malignancy [7]. It also has a distinctive role in the progression of non-small cell lung carcinoma (NSCLC) [9]. Finally, various forms of human cancers demonstrate amplification of the PRKCI gene, including esophageal squamous cell carcinoma, non-small cell lung cancer, and ovarian cancer [10,11].

This altered function of PKCɩ can be because of variations in the genome, specifically single nucleotide polymorphisms. SNPs are genetic variants of a single nucleotide at a particular genomic position found in the genome at every 100 to 300 base pairs [12]. Studies suggest that SNPs residing in the non-coding (but functional) region are more likely to be a possible cause of most genetic diseases. A non-coding variant rs7539120 found in the upstream enhancer region of NOS1APs was found to be associated with QT intervals in cardiac diseases. Cardiac function was seen to be affected by the variant in intercalated discs of cardiomyocytes [13,14]. Similarly, the association of non-coding SNPs of MEIS1 was observed with restless legs syndrome [14]. On other hand, variants in BCL11A gene have affected fetal hemoglobin levels [15]. Studies reveal that a putative variant of rs183205964, residing in the 5′ UTR of TYMS, alters the functional E-box that causes reduction in TYMS expression [16]; on other hand, a deletion or insertion (rs16430) in 3′ UTR at 6bp of TYMS can reduce the transcription rate of TYMS [17].

A polymorphism (rs3783553) located in the 3′ UTR of IL1A gene was found to be associated with a decreased risk of hepatocellular carcinoma in the Chinese population. In the same region, insertion of ‘TTCA’ can demolish miR-378 and miR-122 binding sites [18]. The association of variant in the 3′UTR of KRAS gene occupying the complementary site let-7 miRNA (KRAS-LCS6) was checked with HNSCC (head and neck squamous cell carcinoma); though the overall risk of HNSCC was not associated with LCS6, it was observed that KRAS-LCS6 carriers had a reduced survival rate compared to others [19]. A single nucleotide polymorphism, rs546950 in PRKCI, was confirmed to be associated with prostate cancer by decreasing the risk of disease in the Iranian population [20]. In our previous study, non-synonymous SNPs of PRKCI were characterized, out of which nine variants, F66Y, G34W, R130H, R127K, R130C, Y169H, G165E, G581V and G398S, were associated with altered structure and function of the protein [21]. 

The objective of the current study was to analyze UTR SNPs in PRKCI through in silico tools because of the significance of UTR variants in many studies and association with various diseases. The aim was to determine the functional importance of SNPs and predict the probable impact of 3′ and 5′ UTR variants on formation or disruption of miRNA binding sites and association with single tissue expression quantitative trait loci (eQTL). This study also aimed to investigate the influence of PRKCI UTR SNPs on mRNA structure and transcription factor binding sites. The genetic intolerance of the PRKCI gene to functional variation was also determined. 

## 2. Methods

### 2.1. Retrieval of UTR SNPs and Annotation Using RegulomeDB

Ensemble genome browser was accessed for PRKCI transcript, ENST00000295797.5 on 15 July 2022 for retrieval of 3′ and 5′ UTR SNPs. Functional annotation and scoring of UTR variants was performed through RegulomeDB (https://regulomedb.org/regulome-search accessed on 10 October 2022). RegulomeDB identifies regulatory roles of non-coding SNPs by integrating high throughput data sets from ENCODE, GEO, and other sources. The function of RegulomeDB is to assign scores to SNPs so that the functional SNPs can be differentiated from a broad pool [22]. RegulomeDB scores of PRKCI non-coding (UTR) variants were obtained by entering rs IDs of individual SNPs. The UTR variants were then analyzed via various bioinformatic tools (Figure 1). 

### 2.2. Assessment of Effect of 3′ UTR SNPs on miRNA Binding Sites

The SNPs were assessed for their association with miRNA using three tools: polymorphism in miRNAs and their target sites (PolymiRTS), miRNASNP, and MicroSNIper databases. PolymiRTS database integrates data from hybrid sequences, crosslink experiments, and ligation to miRNA interactions. It can identify the location of non-coding SNPs whether they are in miRNA target sites or seed regions of miRNA [23]. For analyzing the consequences of non-coding SNPs, the rs ID of each variant was submitted to PolymiRTS database (https://compbio.uthsc.edu/miRSNP/miRSNP_detail_all.php accessed on 10 October 2022). 

For prediction of loss and gain function of SNPs that are in mature miRNA, miRNA target sequences, pre-miRNA, and flanking sequences, miRNASNP was employed. A list of targets was produced with SNP-miRNA/duplexes, energy change and gain/loss caused by SNP in 3′ UTR of gene. miRNASNP is available at (http://bioinfo.life.hust.edu.cn/miRNASNP/#!/ accessed on 10 October 2022) [24]. MicroSNIper program was also used to determine the SNP and miRNA interaction, it employs FASTA alignment program for determination of alteration in 3′ UTR (in a nucleotide) that can change the binding capacity of miRNA on the bases of Watson–Crick matches [25] (http://vm24141.virt.gwdg.de/services/microsniper/ accessed on 10 October 2022).

### 2.3. Effect of UTR SNPs on Tissue Expression Using eQTL Analysis

For determination of association between mutations and genetic expression (human tissues), Genotype-Tissue Expression (GTEx) was used. GTEx finds relationship of these traits with the diseases [26] (https://gtexportal.org/home/ accessed on 10 October 2022). It was accessed by submitting individual variants’ rs IDs to find consequences of UTR SNPs.

### 2.4. Determination of Effect of SNPs on Secondary Structure of mRNA

The SNPs annotated as 2a and 2b after regulome DB analysis, variants having association with miRNA, and those having expression in tissues were selected for further analysis. A total of 54 SNPs were selected. Using the RNAfold server, the minimum free energies (MFE) and mRNA secondary structures of wild-type and SNPs were obtained through Vienna RNA package. This package is based on minimum free energy algorithms [27]. The results were obtained using wild-type and SNPs sequences that were obtained from Ensembl.

### 2.5. Analysis of 5′ UTR SNPs on Transcription Factor Binding Sites (TFBS)

The impact of 5′ UTR variants on transcription factor binding sites was evaluated through PROMO. It is a virtual laboratory for determination of putative TFBS in sequences from human or other species. It is linked to TRANSFAC database and uses TFBS defined by it for construction of specific binding site matrices [28]. Both wild and mutated sequences of SNPs were used as input data for obtainment of results. 

### 2.6. Variation Tolerance in PRKCI

The human gene database GeneCards was employed for obtainment of human gene damage index (GDI) and residual variation tolerance score of PRKCI gene. The haploinsufficiency scores, loss of function observed to expected (o:e) ratio, and loss of function intolerance probability (pLI) scores were retrieved from DECIPHER and gnomAD [29]. 

## 3. Results

### 3.1. Retrieval of UTR SNPs of PRKCI from Ensembl and Scoring on RegulomeDB

A total of 713 UTR SNPs including 597 3′ UTR and 116 5′ UTR were acquired from Ensembl data base (Figure 2a) (Appendix A). To check whether the SNPs are pathogenic, likely pathogenic, or acting as a risk factor for a disease, the retrieved UTR SNPs were subjected to functional evaluation through the RegulomeDB server. Out of all SNPs, 556 3′ UTR-located and 109 5′ UTR-located were found on RegulomeDB; all of them distributed in Figure 2a (3, 3′ UTR and 3, 5′ UTR), 2b (25, 5′ UTR), 3a (38 3′ UTR), 4 (29 3′ UTR and 81 5′ UTR), 5 (210 3′ UTR), 6 (178 3′ UTR), or 7 (98 3′ UTR) ranks. With increasing ranks, evidence of transcription factor binding increases (Figure 2b) (Appendix A). A sum of thirty-one UTR SNPs with three in 3′ UTR and twenty-nine in 5′ UTR region were annotated as ≤2b by RegulomeDB having variable RegulomeDB scores are mentioned in Figure 2c. 

### 3.2. Association of UTR SNPs with miRNAs

The impact of 3′ UTR SNPs acquired from Ensembl on miRNA was determined by employing PolymiRTS, miRNASNP, and MicroSNIper. The purpose of using PolymiRTS was to look for SNPs that have impact on miRNA target sites by creating or disrupting miRNA seed regions. The conservation scores, ancestral alleles, changed alleles, context, and score changes are provided in Table 1 The function of an SNP at the PolymiRTS site has been described in four categories. D stands for disruption of conserved site of miRNA, N is for non-conserved miRNA disruption, the formation of new miRNA is denoted by C, and O is used when ancestral alleles are not determined. PolymiRTS determined the association of 31 variants with miRNAs along with disruption scores. Twenty-five SNPs were observed to disrupt the conserved site of miRNA with scores ranging 2–10. Certain alleles from twenty-seven SNPs were found to disrupt the non-conserved miRNA with a score range of 0–16. Only one SNP, rs9824427, was involved in formation of new miRNA (Appendix A). 

miRNASNP determined the number of losses and gains of miRNAs along with linkage disequilibrium. According to miRNASNP, 33 SNPs were involved in the loss of miRNA target sites, while 35 SNPs were causing gain of targets sites. A total of 29 SNPs were altering miRNA by both gain and loss of target sites. Linkage disequilibrium of 17 variants was also confirmed by miRNASNP (Appendix A). MicroSNIper also determined association of 36 SNPs with miRNA (Appendix A).

The combined result from these three tools indicated that twenty-three SNPs are functionally important, having the highest likelihood of altering the miRNA target sequence. The miRNAs that were overlapping in all tools are in bold text (Appendix A). 

### 3.3. Determination of UTR SNPs eQTLs

All the 3′ and 5′ UTR SNPs obtained from ENSEMBL were analyzed on the GTEx portal for single tissue eQTL analysis. Out of 697 3′ UTR and 116 5′ UTR SNPs, only two SNPs were found on the GTEx portal. Both the SNPs are located in the 3′ UTR region. Significant association was found between PRKCI expression in stomach and esophagus mucosa and SNP rs140672226 with *p*-values 0.000064 and 0.000099, respectively (Figure 3a,b). SNP rs2650220 was found to be significantly associated with expression in esophagus mucosa (*p* = 7.9 × 10^−7^) (Figure 3c) (Table 2). 

### 3.4. Impact of UTR SNPs on Secondary Structure of mRNA

A total of 54 functionally significant UTR SNPs from RegulomeDB (rank ≤ 2b), SNPs having association with miRNA, and SNPs having tissue-based expression found via GTEx analyses were subjected to further analysis through RNAfold to check their impact on mRNA structure. RNAfold analysis revealed alteration in minimum free energy (MFE) and mRNA structure of wild-type and mutant-type by 35 SNPs while the remaining SNPs did not cause any substantial change in MFE and structure of mRNA. Out of 35 SNPs, 15 were in the 3′ UTR while 20 were located in the 5′ UTR (Table 3) (Figure 4 and Figure 5). 

For 3′ UTR SNPS, a significant change in energy was observed in rs1447651774 and rs115170199. The change in energy was −17.70 (kcal/mol) to −13.60 (kcal/mol) in rs1447651774 while in rs115170199 ∆G = −8.30 (kcal/mol) to −3.10 (kcal/mol) (Table 4). There were also obvious changes in the structure of the mRNA; SNP rs1447651774 caused an increase in loop size on the lower side of the mRNA, while rs115170199 totally changed the structure by reducing the size of the loop on the upper side and introduced a new loop in the middle of the mRNA (Figure 4). The increase in energy of wild-type structures suggested that SNPs were destabilizing. The other SNPs were also observed to alter mRNA structure with marginal variation in energy. 

In 5′ UTR-located SNPs, considerable changes in ∆G were seen in four SNPs. The ∆G for rs778557075, rs968409340, and rs750297755 was from −22.70 to −18.20 (kcal/mol), 19.50 to −15.20 (kcal/mol), and −19.20 to −14.10 (kcal/mol), respectively. The energy is increased, hence destabilizing the mRNA structure (Table 3). A noteworthy change in mutant structures is observed especially in the case of rs750297755 (Figure 5). In SNP rs542458816, ∆G was found to be decreased from −15.70 to −21.50 (kcal/mol). This signifies the SNP as stabilizing for mRNA structure.

### 3.5. Effect of 5′ UTR SNPs on Transcription Binding Factors

A total of 29 5′ UTR SNPs annotated as 2a and 2b were selected for transcription factor binding analysis. Wild-type and mutated sequences of SNPs were used as input parameters for PROMO, which is a virtual library for the analysis of putative transcription binding factors. Fourteen SNPs were found to affect transcription factor binding sites (TFBS) by addition or removal of binding sites as well as arrangement of these sites on DNA sequence (Figure 6 and Figure 7).

In SNP rs994884642, the wild-type sequence has nine predicted TFBS in total, whereas in comparison, the mutant sequence has eight TFBS with loss of Egr-3 in the mutant type. It is evident from graphical representation that the number and arrangement of binding sites on the gene has been altered between wild-type and mutant forms (Figure 6a, Appendix A). In the case of SNP rs1482898617, the wild-type (WT) has nine TFBS and the mutant type (MT) has eight TFBS, having Pax-5 absent in case of variant. Here too, there is a rearrangement in existing TFBS locations between wild-type and mutant (Figure 6b, Appendix A).

Wild-type TFBS are eight in SNP rs1024270582 while in mutant-type the number has been reduced to six TFBS with the loss of GCF and WT1 (Figure 6c, Appendix A). TFBS in rs1031689697 wild-type are eight; on other hand, TFBS in mutant-type are six, having GCF and WT1 removed as in the case of rs1024270582. The result illustrates a substantial change in TFBS caused by SNP (Figure 6d, Appendix A).

For rs1366285245, there is a loss of transcription binding factor WT1 in the mutant-type, reducing the number from nine (in wild type) to eight in the mutant-type, highlighting the fact that the TBFS are changing (Figure 5, Appendix A). The WT of rs111868479 contains 10 TBFS, and while the MT contains 8 TBFS, in the mutant, one transcription factor ETF is missing that will affect the process of transcription. (Figure 6f, Appendix A). 

Transcription factor binding site analysis of SNP rs763083643 WT showed that TFBS are predicted to be 10 while for MT, TFBS are 7 with ETF, E2E-1, and WT1 removed. The reduction in transcription factors from 10 to 7 will most probably affect transcription significantly (Figure 7a, Appendix A). rs766667625 WT has 10 TFBS; while MT has 8, here, 2 TFBS ETF and E2F-1 are missing in the mutant, which will probably change the process (Figure 7b, Appendix A). The TFBS in rs778557075 WT are nine, and in WT TFBS are seven, in which AR-alpha in the wild-type was replaced by XBP-1 and two TFBS, WT1 and E2F-1, were lost (Figure 7c, Appendix A). 

The SNP rs968409340 WT contains nine TFBS and WT contains six from which MAZ and WT1 are removed. The change from nine to six can lead to alteration in transcription, subsequently resulting in faulty protein (Figure 7d, Appendix A). For WT rs750297755, there are 10 TFGS: 0 = C/EBPbeta, 1 = GR-alpha, 2 = AR-2alpha, 3 = XBP-1, 4 = RXR-alpha, 5 = PPAR-alpha, 6 = TFII-I, 7 = MAZ, 8 = WT1, 9 = E2F-1, 10 = FOXP3, and for MT there are eight having loss of WT1 and MAZ (Figure 7e, Appendix A). In the case of SNP rs542458816, the WT has 10 TFGS and WT has 14 TFGS, with the addition of TFBS AhR-Amt, E2E-1, ETF, and FOXP3 (Figure 7f, Appendix A). Transcription factors in this case are increased from 10 to 14 and the arrangement is also altered.

Out of all two SNPs, rs542458816 and s763083643 are examined to cause a considerable change in transcription factors binding sites, with rs542458816 increasing TFBS from 10 to 14 and with rs763083643 decreasing from 10 to 7. 

### 3.6. Variation Tolerance in PRKCI

Tolerance of genes towards genetic variations can be examined using certain parameters. Tolerance of genetic variation in PRKCI was analyzed by using five types of scores that include gene damage index score (GDI), probability of loss of function intolerance (pLI), residual variation tolerance percentage, haploinsufficiency score, and observed to expected loss of function ratio (o:e). The gene is regarded as haploinsufficient if o:e is near to 0.1 and is considered as tolerant if the o:e value is close to 1 [30]. The o:e value for PRKCI was found to be 0.26, indicating its tendency towards haploinsufficiency. The pLI (probability of LoF intolerance) is the probable haploinsufficiency of the target gene. Genes with pLI values ≥ 0.9 are intolerant to variants while those with pLI values ≤ 0.1 are tolerant to loss of function variants [22]. The pLI values for PRKCI fall in the tolerant range. Genes with 25% tolerance percentile score and below are regarded as intolerant to variation [23]. PRKCI, by its 17.6% percentile value, is regarded as intolerant (Table 4). Low GDI and high haploinsufficiency values suggest that loss of function variants in PRKCI can be disease-causing and intolerant.

## 4. Discussion

SNPs are identified as the variations in genes that have large occurrence and proportion compared to other variations. SNPs, particularly, in the coding region have direct association with phenotype change [31,32]. However, in comparison to coding regions, most SNPs are located in the non-coding regions through GWAS [33]. In the case of cancer non-coding SNPs, localization within the UTRs is considered to be especially damaging. The identified variations in these regions are found to perturb the structure of miRNA and thereby the interactions with other miRNAs and gene products [34,35,36]. UTRs maintain post-translation regulation in gene expression, any variation in the UTR region can be linked to severe pathologies [37,38]. In the present study, a total of 597 3′ UTR and 116 5′ UTR were retrieved from Ensembl. After subjecting them to RegulomeDB analysis, it was found that 6 SNPs were annotated as 2a and 25 were labelled as 2b with the remaining SNPs distributed from categories 3a to 6. In one study carried out on the Korean population, SNPs within the XRCC1 gene were identified and linked with lung cancer. In this study, 10 SNPs were found to have functional relevance, one SNP (rs2854509) fell into the score range of 1a while one SNP (rs2682563) fell into the score range of 1d and eight SNPs were found to have scores of 1f. These 10 SNPs were directly involved in affecting XRCC1 expression [39]. In another study, SNPs within non-coding variants specifically involved in coronary artery diseases were identified. A total of 1200 SNPs were tested, out of which 858 turned out to have scores ranging 1–6. Moreover, 97 out of 858 had scores less than 3 [40].

Different bioinformatics tools and software introduced by the researchers have paved the way to analyzing SNPs in the non-coding region, which seemed a difficult task only a few years back. PolymiRTS, miRNASNP, and MicroSNIper are such tools available online that help in the determination of SNP association with miRNA. The tools were employed for the identification of alteration in miRNA sites of PRKCI by UTR SNPs [41]. It is worthy of noting here that every SNP has an independent, unique effect on different miRNAs; for example, if an SNP is linked to five miRNAs, it will have four different effects. The collective result from these tools suggested twenty-three SNPs that can alter microRNA targets of the sequence. MicroRNAs influence the stability of mRNAs, which can affect protein outcomes. MicroRNAs also widely function in various regulatory pathways, which can be effected by damaged microRNAs, leading to pathologies [42]. 

In a study, variations in the 3’ UTR of miRNAs that target the APOB gene were identified using the PolymiRTS tool in which the same mutation at the same site with the SNP ID rs72654430 (T/C) was identified in the two miRNAs; namely, hsa-miR-29a-5p and hsa-miR-378a. Moreover, SNP ID rs72654430 was identified to disturb the conserved site of 3’ UTR site of hsa-miR-29a-5p with a context score of −0.138, while in the other functional class of miRNA, hsa-miR-378a-5p, the SNP ID rs72654430 created a new miRNA site, and the score was identified as −0.242. The higher context score was found to be associated with greater structural change and disruption within miRNA [43]. In another study, the gene-to-gene interactions and expression of FOXC2 were found to be severely affected with the mutations at 5′ and 3′ sites within miRNAs that targeted it. Mainly, the three miRNAs identified as hsa-miR-6886-5p, hsa-miRS-6886-5p, and hsa-miR-6720-3p were profoundly affected by the SNPs with IDs, rs201118690, rs6413505, and rs201914560, respectively [44]. 

All the 5′ and 3′ UTR SNPs were analyzed on GTEx portal for association with tissue qQTLs. Only two SNPs residing in 3′ UTR were found to be associated with tissue eQTLs. SNP rs140672226 was linked with expression in stomach and esophagus mucosa with *p*-values 0.000064 and 0.000099, respectively (Figure 2a,b), while SNP rs2650220 was found to be related with expression in esophagus mucosa significantly. In a paper, LINC00461 expression, particularly in the hippocampus, was found to augment schizophrenia onset [45]. GTEx analysis was also used in one study to profile 14,100 lncRNA expressions in 49 different tissues to discover the association of lncRNA with diabetes, coronary disease, and other related diseases [46].

The impact of functionally important SNPs from RegulomeDB, miRNA association and GTEx analysis (total 54) on structure on mRNA was determined using RNAfold. Two SNPs from 3′ UTR (rs1447651774, rs115170199) and three from 5′ UTR (rs778557075, rs968409340, 750297755) were destabilizing the overall structure of mRNA with substantial energy change. Studies show that control and regulation of mRNA structure stability is important for proper protein output [47]. In a study, SNP rs929271 leukemia inhibitory factor (LIF) was found to destabilize the mRNA structure with an energy change from −18.47 and −18.17 kcal/mol using the RNAfold server [48]. In another research paper, rs2229611 SNP in G6PC1 associated with glycogen storage by affecting mRNA stability, RNAfold also predicted the destabilizing effect of this SNP on mRNA [49]. The 5ʹUTR SNPs annotated as 2a and 2b were analyzed on PROMO for their impact on creation or interruption of DNA binding sites for transcription factors. SNPs rs542458816 and s763083643 were observed to alter transcription factors binding sites majorly. The SNP rs750297755 is involved in destabilizing the mRNA structure, revealing its likelihood of association with disorders. A paper illustrated the effect of SNPs that are located at the osteopontin promoter on transcription binding sites in which SNP rs2728127 was decreasing the number of transcription factors [50]. 

Genetic mutation intolerance of PRKCI in terms of GDI, haploinsufficiency and o:e suggested that PRKCI is likely to be intolerant to functional SNPs that can cause damage. Human gene damage can be measured using GDI (gene damage index), which is the metrics of mutational destruction at gene level [51]. This study highlighted the potential significance of 3′ and 5′ UTR SNPs and their regulatory effects on gene expression. However, there are some limitations of the current study, such the bioinformatic study performed on UTR variants of PRKCI not being disease-specific. A gene can be involved in the causation of various diseases, and variant analysis of a gene performed with respect to a disease model can help in determining the significance of gene variants in a disease. The current study includes only UTR variants, but analysis of other variant types can also provide insight into the pathogenicity of the PRKCI gene. The functional significance of UTR variants determined via in silico analysis requires validation through experimentation. Results after in vitro and in vivo validation in future could help in the diagnosis and therapeutics of various diseases. 

## 5. Conclusions

The role of UTRs cannot be denied in gene expression; they are crucial to post-translational regulation. The alterations in this region can be damaging and are linked with various serious diseases. In this study, for the first-time, a comprehensive assessment of 3′ and 5′ UTR SNPs in PRKCI was performed using bioinformatics tools. Twenty-three SNPs were found to disrupt miRNA, thus affecting gene expression, while five SNPs (rs1447651774 and rs115170199 from 3′ UTR and rs778557075, rs968409340 and 750297755 from 5′ UTR region) were predicted to have detrimental effects on the structure of the mRNA, thus influencing the overall functional consequences of the protein. Along with destabilizing the mRNA structure, the SNP rs542458816 is seen to effect transcription factor binding sites the most. The results need experimental confirmation, so that they can provide direction to the development of specific disease diagnostics and treatment interventions by targeting functionally important SNPs.

## Figures and Tables

**Figure 1 genes-14-00247-f001:**
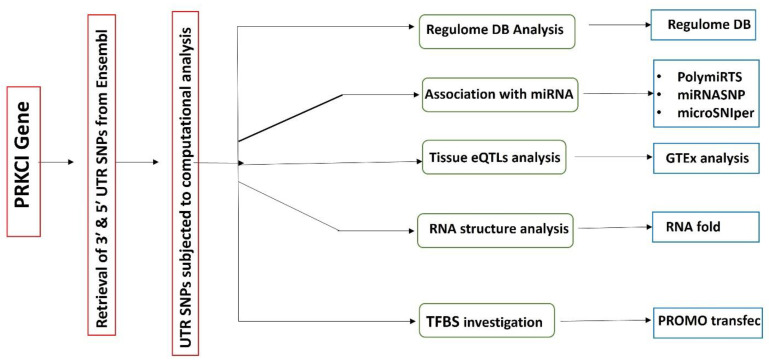
Flowsheet of methods involved in study.

**Figure 2 genes-14-00247-f002:**
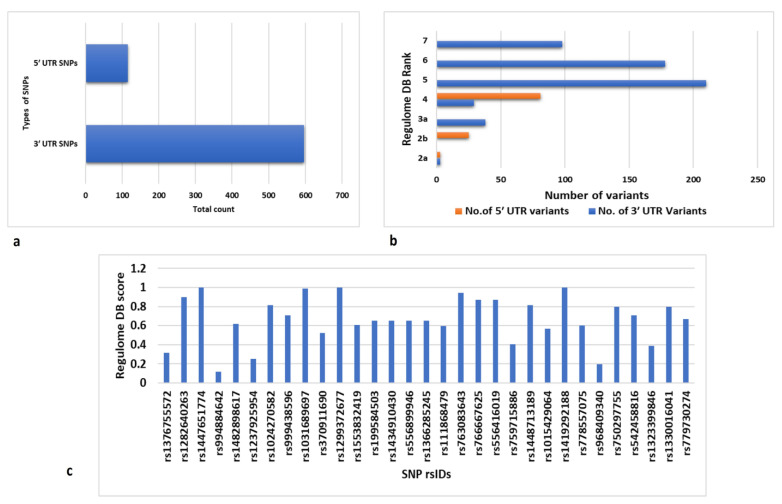
(**a**) Total count of 3′ and 5′ UTR variants of PRKCI recovered from Ensembl; (**b**) annotation and scoring of UTR SNPs obtained through RegulomeDB; (**c**) RegulomeDB scores of SNPs annotated as 2a and 2b.

**Figure 3 genes-14-00247-f003:**
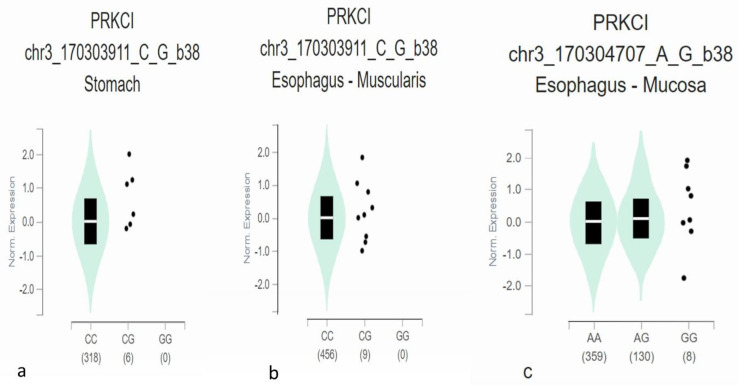
(**a**,**b**) Violin plots of SNP rs140672226 and (**c**) rs2650220 for single tissue eQTLs analysis through GTEx.

**Figure 4 genes-14-00247-f004:**
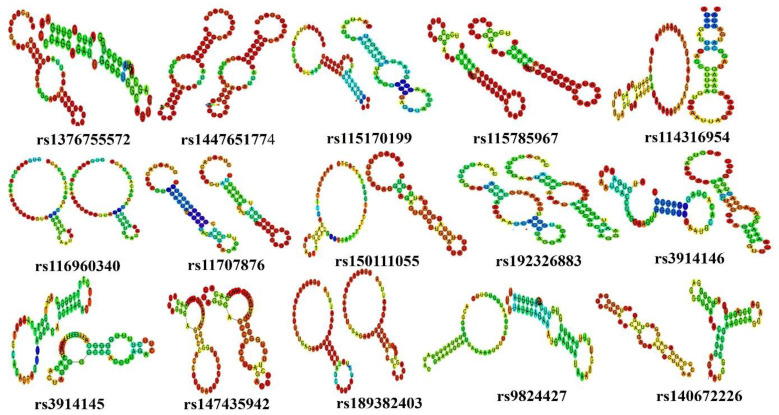
Comparison of structures of wild and mutated mRNA of 3′ UTR SNPs through RNAfold.

**Figure 5 genes-14-00247-f005:**
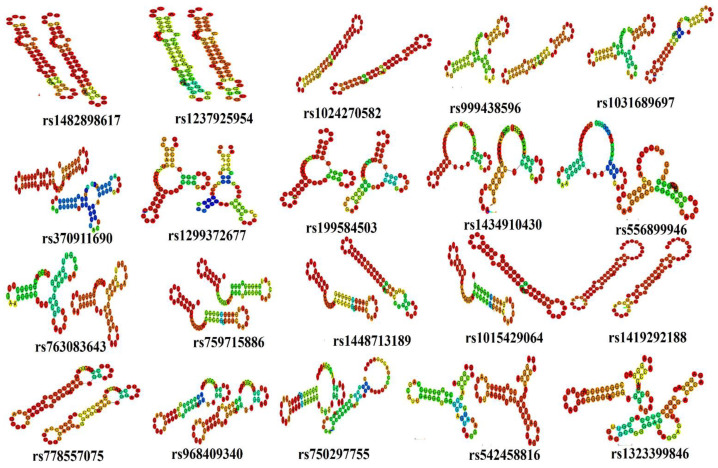
Comparison of structures of wild and mutated mRNA of 5′ UTR SNPs through RNA fold.

**Figure 6 genes-14-00247-f006:**
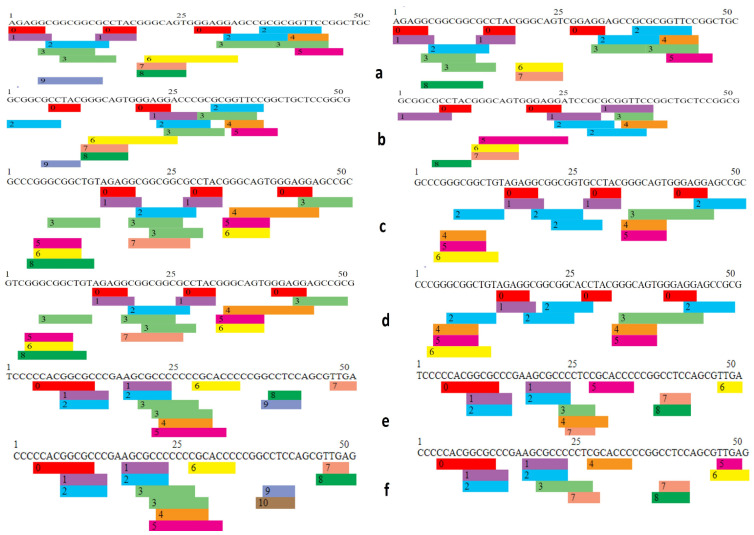
Comparison of transcription factor binding sites of wild and mutated sequences: (**a**) rs994884642: the wild sequence has 9 transcription factors in total while the mutant has 8 TBFS; (**b**) rs1482898617: the wild type (WT) has 9 while Mutant type (MT) has 8; (**c**) in rs1024270582 WT, TBFS are 8 while in MT the number is 6; (**d**) for TBFS in rs1031689697 WT, there are 9 TFBS, there are, while in MT there are 6; (**e**) for rs1366285245 WT the TBFS are 9 and for MT TBFS are 8; (**f**) the WT of rs111868479 contain 10 TBFS and in MT the number is reduced to 8.

**Figure 7 genes-14-00247-f007:**
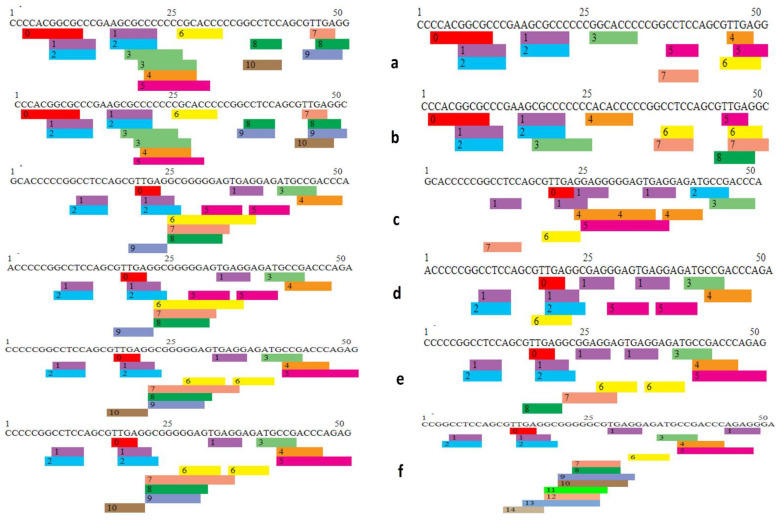
Transcription factor binding sites analysis of SNPs. (**a**) For rs763083643, wild-type (WT) has 10 while mutant-type (MT) has 7 TFBS. (**b**) rs766667625 WT has 10 TFBS while MT has 8. (**c**) The TFBS in rs778557075 WT are 9 in MT TFBS are 7. (**d**) rs968409340 WT contains 9 TFBS (**e**) For WT rs750297755 there are 10 TFBS, and for MT there are 8. (**f**) rs542458816 WT has 10 TFBS and MT has 14 TFBS.

**Table 1 genes-14-00247-t001:** Minimum free energy (MFE) of wild and mutant structures of mRNAs caused by 5′ UTR SNPs.

SNPs	Location	MFE Wild (kcal/mol)	MFE Mutated (kcal/mol)
rs1482898617(C/T)	3:170222490	−24.90	−23.90
rs1237925954 (C/T)	3:170222491	−22.60	−24.80
rs1024270582 (C/T)	3:170222468	−26.20	−27.40
rs999438596 (G/A)	3:170222469	−21.60	−21.20
rs1031689697 (C/T)	3:170222470	−16.20	−19.90
rs370911690 (C/G)	3:170222471	−21.60	−22.00
rs1299372677(C/T)	3:170222621	−14.30	−12.40
rs199584503 (C/T)	3:170222623	−14.50	−11.80
rs1434910430 (C/G)	3:170222624	−10.60	−10.40
rs556899946 (C/G)	3:170222626	−10.60	−12.30
rs763083643 (C/T)	3:170222629	−13.40	−16.40
rs759715886 (C/A)	3:170222634	−23.70	−23.40
rs1448713189 (C/T)	3:170222635	−26.90	−26.40
rs1015429064 (C/T)	3:170222636	−24.20	−26.60
rs1419292188 (G/T)	3:170222653	−25.60	−19.00
rs778557075(C/A)	3:170222655	−22.70	−18.20
rs968409340(G/A)	3:170222657	−19.50	−15.20
rs750297755 (G/A)	3:170222658	−19.20	−14.10
rs542458816 (A/C)	3:170222661	−15.70	−21.50
rs1323399846 (G/C)	3:170222662	−15.00	−16.00

**Table 2 genes-14-00247-t002:** Single tissue eQTL association of UTR variants taken from GTEx portal with NES and *p*-values.

Variant ID	Region	SNP ID	*p*-Value	NES	Tissue eQTL
chr3_170303911_C_G_b38	3′ UTR	rs140672226	0.000064	0.82	Stomach
			0.000099	0.49	Esophagus—muscularis
chr3_170304707_A_G_b38	3′ UTR	rs2650220	7.9 × 10^−7^	0.15	Esophagus—mucosa

**Table 3 genes-14-00247-t003:** Genetic variation tolerance of PRKCI with human gene damage index score (GDI), residual variation tolerance score, haploinsufficiency score (%HI), observed to expected (o:e) ratio, and loss of function intolerance probability (pLI) values.

Gene	GDI Value	Residual Variation Tolerance Score	Haploinsufficiency Score	Loss of Function (o:e) Ratio	pLI
PRKCI	0.99	17.6%	19.34	0.26 (0.16–0.44)	0.07

**Table 4 genes-14-00247-t004:** Minimum free energy (MFE) of wild and mutant structures of mRNAs caused by 3′ UTR SNPs.

SNPs	Location	MFE Wild (kcal/mol)	MFE Mutated (kcal/mol)
rs1376755572(T/C)	3:170303934	−13.60	−11.20
rs1447651774(G/A)	3:170303943	−17.70	−13.60
rs115170199 (C/G)	3:170303662	−8.30	−3.10
rs115785967 (A/G)	3:170304156	−13.90	−13.20
rs114316954 (C/T)	3:170304522	−3.10	−1.20
rs116960340 (A/G)	3:170304540	−2.50	−2.10
rs11707876 (G/C)	3:170304653	−5.20	−1.40
rs150111055 (C/T)	3:170304667	−4.50	−5.00
rs192326883 (C/T)	3:170305221	−4.50	−4.60
rs3914146 (G/T)	3:170305232	−5.10	−5.50
rs3914145 (A/G)	3:170305253	−7.40	−7.30
rs147435942 (A/T)	3:170305263	−8.20	−6.80
rs189382403 (A/C)	3:170305381	−7.00	−6.10
rs9824427 (C/A)	3:170305608	−6.00	−3.20
rs140672226 (C/G)	3:170303911	−9.90	−8.40

## Data Availability

All the relevant data is provided in the manuscript and any Appendix A used and/or analyzed during the current study are available from the corresponding author upon reasonable request.

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
