# Peer review of "Investigation of UTR Variants by Computational Approaches Reveal Their Functional Significance in PRKCI Gene Regulation"

_genes, 2023, doi:10.3390/genes14020247_

Round 1

Reviewer 1 Report

The manuscript entitled: Investigation of UTR variants by computational approaches re-veal their functional significance in PRKCI gene regulation has been reviewed. 

Major Queries/Comments:

1- Statistical analysis section is missing. Authors should mention the name of the statistical software used, the name of the tests used, the significance level of the P-value, etc...

2- Please modify figures 5 and 6 to make them more precise. 

3- The limitation section is missing in the discussion.

4- There paper lacks experimental confirmation to improve the results.

Author Response

We have addressed all the comments raised by reviewers. Following are the answers

  • Reviewer1

The manuscript entitled: Investigation of UTR variants by computational approaches re-veal their functional significance in PRKCI gene regulation has been reviewed. 

  • The statistical analysis section is missing. Authors should mention the name of the statistical software used, the name of the tests used, the significance level of the P-value, etc...

Response: We thank reviewer for the comment. Most of the computational tools used for variant analysis have different algorithms and a scoring or cut off criterion of their own based on which SNPs are annotated and considered as functionally substantial. In many cases, most of the tools do not require separate statistical analysis for validation of results. Some tools like GTEx calculate p-values and significance through their own algorithms. They are some studies in which various in silico approaches have been employed without applying any external statistical tests    (DOI: 10.7717/peerj.7667, PMID: 31592138), (DOI: 10.1016/j.dib.2017.07.057, PMID: 28856185), (DOI: 10.1016/j.compbiomed.2021.104986, PMID: 34739970),  (DOI: 10.2174/1871524919666191014104843, PMID: 31660846), and (DOI: 10.3390/biom11111733, PMID: 34827731).

  • Please modify figures 5 and 6 to make them more precise. 

 Response: Figures 5 and 6 are re-labeled as Figures 6 and 7 highlight the number and position of transcription factor binding sites on the wild and mutated genome. The transcription factor binding sites' details are provided in supplementary data file 4. The analysis was performed on PROMO, an online tool for determining putative TFBS in sequences from humans or other species. It is linked to the TRANSFAC database and generates the images. The illustrations used in the study for 3ʹ and 5ʹ UTR variants were downloaded from PROMO directly and were provided in Figures 6 and 7, that’s why it will not be possible to re-settle the illustrations manually. Any changes made can disrupt the number and arrangement of TBFS.

  • The limitation section is missing in the discussion.

    Response: The limitation of the study is included in the discussion section

 “However, there are some limitations of the current study, like the bioinformatic study performed on UTR variants of PRKCI is not disease specific. A gene can be involved in causation of various diseases, variant analysis of a gene performed with respect to a disease model can help in the significance of gene variants in the disease. The current study includes only UTR variants; analysis of other variant types can also give insight into the pathogenicity of the PRKCI gene. The functional significance of UTR variants determined via in silico analysis requires validation through experimentation. Results after in vitro and in vivo validation in the future could help in diagnosis and therapeutics of various diseases”.

  • The paper lacks experimental confirmation to improve the results.

Response: The focus of the current study is on identification of functionally significant UTR variants in the PRKCI gene and their effect on the regulation of gene expression through computational approaches.  The experimental validation of the study is a part of our future objective in which we will be confirming the association of variants with miRNA and the impact of variants on post-translational regulation and mRNA secondary structure.

Reviewer 2 Report

Gist/Summary: The authirs identufy single nucleotide polymorphisms (SNPs) in the unyranslated regions of PRKCI gene using myriad of bioinformatics tools and check for the genetic intolerance and they plausibly suggest the functional importance of the PRKCI gene. The manuscript is just okay but used a large number of Ifs and buts which could be improved many ways. I have made subtle suggestions in the attached document

There must be a pictorial methodology

The background is absymally less and must be rewritten with a strong rationale

What was the essence of thes etools, for example genetic intolerance using genecards may not be a good idea.

Were there any SNPs attributed to lncRNAS or further upstream?

any specific regulatory mechanisms through ehancers, CTCF motifs?

The figure 4 could be a table isntead with Ref/Alt SNP location an dten rs ids.

The figures 2 and 3 could be improved with high resolution. 

Scores on a scale of 0-5 with 5 being the best 

Language: 3

Novelty: 2

Brevity: 4.5

scope/Relevance: 4

Author Response

Dear Editor

We have addressed all the comments raised by reviewers. The following are the answers.

Reviewer 2

  • There must be a pictorial methodology

Response: We thank the reviewer for the suggestion. Pictorial methodology representing all the methods and tools has been added as figure 1 to the methods section of the manuscript on page 5.

  • The background is abysmally less and must be rewritten with a strong rationale.

 Response: The background of the study has been modified with relevant information, specifically about non-coding and UTR variants. The last section of the introduction has been rewritten with a focused rationale added to pages 4 and 5 of the manuscript.

  • What was the essence of these tools, for example, genetic intolerance using genecards may not be a good idea?

Response: The use of computational tools provides a platform for screening deleterious or functionally important variants in a short time, that can be further used in in vitro and in vivo studies. Gene Cards is a comprehensive database that gives genetic, transcriptomic, proteomic, and genomic data.  Further, the confirmation of variation intolerance was done via loss of function probability (pLI), haploinsufficiency scores, and loss of function observed to the expected ratio which was obtained from DECIPHER and gnomAD. The use of tool helped us in gaining insight of mutation susceptibility of the gene.

  • Were there any SNPs attributed to lncRNAS or further upstream?

 Response: Thank you for the comment. The SNPs attributed to lncRNAs were assessed via “lncRNASNP2” which is a web-based tool for the determination of single nucleotide polymorphisms in human lncRNAs. After analysis of all the filtered 3ʹ and 5ʹ UTR SNPs on lncRNASNP2, no SNP was found to affect lncRNA or miRNA-lncRNA binding.

  • Any specific regulatory mechanisms through enhancers, CTCF motifs?

Response: The tools used in the study highlighted the change in the number and arrangement of transcription factor binding sites of the genome induced by SNPs. Unfortunately, it did not provide any specific mechanism regarding the change in function of transcriptional activators or repressors, nor did it give any idea about how the SNPs can affect transcription via promoters, enhancers, and CTCT motifs. The research of in-depth mechanisms of influence of SNPs on gene transcription can be a goal for future studies.

  • Figure 4 could be a table instead with Ref/Alt SNP location and rs ids.

Response: We thank the reviewer for the comment. Figure 4, which is renamed as figure 5 displays the effect of 5ʹ UTR variants on mRNA structure with both wild-type and mutant-type structures. However, following the reviewer’s advice, the location of both 3ʹ and 5ʹ UTR variants have been added to tables 2 and 3 against their rs IDs and MFE change.

  • Figures 2 and 3 could be improved with high resolution. 

Response: Thank you for the comment. Figure 2 and 3 which are re-labelled as figure 3 and 4 have been replaced with better resolution figures on page 11 and 13 respectively in the result section of the manuscript.

Round 2

Reviewer 1 Report

The authors have addressed all my comments.

Reviewer 2 Report

I am convnced with all changes rendered But I would appreciate if the authors corrected the bludners in language.  Pl see the attached
